# Galacto-Oligosaccharide RP-G28 Improves Multiple Clinical Outcomes in Lactose-Intolerant Patients

**DOI:** 10.3390/nu12041058

**Published:** 2020-04-10

**Authors:** William Chey, William Sandborn, Andrew J. Ritter, Howard Foyt, M. Andrea Azcarate-Peril, Dennis A. Savaiano

**Affiliations:** 1Gastroenterology and Nutrition Sciences 3912 Taubman Center, SPC 5362, Ann Arbor, MI 48109, USA; 2Division of Gastroenterology, University of California San Diego, San Diego, CA 92093, USA; wsandborn@ucsd.edu; 3Ritter Pharmaceuticals, Inc., Los Angeles, CA 90067, USA; andrew@ritterpharma.com (A.J.R.); HFoyt@viacyte.com (H.F.); 4Department of Medicine, Division of Gastroenterology and Hepatology and UNC Microbiome Core, School of Medicine, University of North Carolina at Chapel Hill, NC 27514, USA; azcarate@med.unc.edu; 5Meredith Professor of Nutrition Science, Institution: Purdue University, West Lafayette, IN 47907, USA; savaiano@purdue.edu

**Keywords:** lactose intolerance, lactase non-persistence, galacto-oligosaccharide, gut microbiome, abdominal pain, bloating, gas, diarrhea

## Abstract

**Background and Aims:** Lactose intolerance (LI) is a global problem affecting more than half of the world’s population. An ultra-purified, high-concentration galacto-oligosaccharide, RP-G28, is being developed as a treatment for patients with LI. The efficacy and safety of RP-G28 in reducing symptoms of lactose intolerance were assessed in a blinded, randomized, placebo-controlled trial. **Methods:** In this multiclinical site, double-blinded, placebo-controlled trial, 377 patients with LI were randomized to one of two doses of orally administered RP-G28 or placebo for 30 days. A LI test and symptom assessment were performed at baseline and on day 31. The primary endpoint was a ≥4-point reduction or a score of zero on LI composite score on day 31. Voluntary milk and dairy intake and global outcome measures assessed patients’ overall treatment satisfaction and quality of life before therapy and 30 days after therapy. This study received Institutional Review Board (IRB) approval. **Results:** For the primary endpoint, 40% in the RP-G28 groups reported a ≥4-point reduction or no symptoms on LI symptom composite score compared to 26% with placebo (*p* = 0.016). Treatment with RP-G28 also led to significantly higher levels of milk and dairy intake and significant improvements in global assessments compared to placebo. RP-G28 but not placebo led to significant increases in five *Bifidobacterium* taxa. **Conclusions:** RP-G28 for 30 days significantly reduced symptoms and altered the fecal microbiome in patients with LI. Treatment with RP-G28 also improved milk/dairy consumption and quality of life and was safe and well tolerated.

## 1. Introduction

Non-persistence of the lactase enzyme in the small intestinal mucosa (LNP) affects 65–70% percent of the population globally, impacting health and potentially causing distressing gastrointestinal (GI) symptoms, which are commonly referred to as lactose intolerance (LI) [1]. LNP results in lactose malabsorption, which allows undigested lactose to move into the colon. Fermentation of lactose in the colon can produce carbon dioxide, hydrogen gas, methane and short-chain fatty acids, leading to a range of abdominal and bowel-related symptoms that can include abdominal pain, cramping, discomfort, bloating, distension, flatulence, increased stool frequency and/or loose or watery stools [2].

Patients with LI often avoid dairy products. Prolonged restriction of dairy can result in insufficient dietary calcium and vitamin D, which can, in turn, lead to osteoporosis, osteomalacia, and hypertension [3,4]. The daily calcium intake in patients with LI, ranging from 320 to 388 mg/day, is significantly less than the recommended amount of 1000–1200 mg/day [5,6].

There is accumulating evidence that treatments which alter the gut microbiome can improve a wide range of diseases, either by reversing disease-associated alterations in the microbiome or “dysbiosis” or by modifying the “normal” gut microbiome. One example of this latter strategy includes galacto-oligosaccharides (GOSs), which pass intact through to the colon, where they stimulate the growth of lactose-metabolizing bacteria. Elevated populations of *Lactobacillus* and *Bifidobacterium* enhance β-galactosidase activity and GOS utilization [7,8], resulting in enhanced fermentation of lactose to glucose, galactose, and short-chain fatty acids as well as reduced lactose-derived gas production, which could be beneficial to the symptoms of LI [9].

RP-G28, an ultra-purified, high-concentration (>95%) galacto-oligosaccharide, has been evaluated in a Phase 2a study in 85 patients [2]. RP-G28 reduced abdominal pain in 50 percent of patients after treatment and 30 days post-treatment (*p* = 0.019). Additionally, RP-G28-treated patients were 6 times less likely than patients given placebo to report LI 30 days post-treatment after reintroduction of dairy foods (*p* = 0.039).

The aim of the current study was to further evaluate the efficacy and safety of two doses of RP-G28 in patients with LI in a larger, multicenter, randomized, double-blind, placebo-controlled, parallel group trial.

## 2. Materials and Methods

### 2.1. Study Overview

A multicenter, randomized, double-blind, placebo-controlled, parallel group clinical trial was conducted to determine the efficacy, safety, and tolerability of two doses of RP-G28 in subjects with moderate to severe LI. The study took place between March and October 2016 at 15 investigative centers throughout the U.S. and included a 7 day screening, a 30 day treatment, and a 30 day post-treatment “real-world” observation, during which dairy was re-introduced to patients’ diets (Figure 1).

The clinical trial consisted of 3 phases: a screening phase, a treatment phase and a post-treatment phase. Patients were screened for LI for 7 days prior to study. Patients were then stratified into higher and lower-dose RP-G28 treatment or placebo for a 30 day treatment phase, during which patients did not consume lactose. On day 31, post-treatment, LI symptom assessments were made. Following this, “real-world” dairy intake began, and LI symptoms were assessed over a 30 day period. 

Study patients, investigators, study site staff, the sponsor, the medical monitor and the study monitors were blinded to the treatment during the trial. All sites obtained Institutional Review Board (IRB) approval. This investigation was carried out in accordance with the Declaration of Helsinki of the World Medical Assembly and its revisions, as well as the rules of Good Clinical Practice (GCP) of the United States FDA (Protection of Human Subjects, 21 CFR 50; IRB, 21 CFR 56; and IND, 21 CFR 312). IRB approval for protocol G28-003 was approved on 26 January 2016, with approval of Amendment #1 on 5 May 2016, and Amendment #2 on 17 June 2016. This clinical trial can be found on the clinical trial registry website (www.clinicaltrials.gov), trial number NCT02673749.

The primary and secondary endpoints provided a comprehensive evaluation of the effect of treatment on the symptoms of LI, milk and dairy consumption, quality of life experiences, and fecal microbiome to assess meaningful treatment benefits.

### 2.2. Screening Period

To establish eligibility, a pre-screening questionnaire was administered to confirm patient perception of LI. Eligible patients were required to have a moderate to severe symptom severity score, and a positive Hydrogen Breath Test (HBT) following a standardized in-clinic lactose challenge. The criteria for diagnosis of LI, and the main criteria for inclusion into the study, are provided in Table 1. 

A 5 h HBT at baseline confirmed LNP in patients. Hydrogen, methane, and carbon dioxide concentrations were measured in exhaled breath following a single-blinded lactose challenge three times during the study. Anhydrous food-grade lactose was administered at 0.35/kilogram body weight on day 0 (baseline, in order to confirm that the patient had symptoms of LI for eligibility), on day 31 (end-of-treatment, primary endpoint), and on day 61 (end of post-treatment period). After fasting overnight, patients were assessed for symptoms of LI using a validated symptom questionnaire (LI Symptom Questionnaire) to create a composite symptom score (comprised of abdominal pain, cramping, bloating and gas) at 6 time points starting at 30 minutes and hourly thereafter for 5 h.

Subjects who met the eligibility criteria completed a global assessment questionnaire and a 7 day lactose consumption assessment recall, provided stool samples, and were randomized to 1 of 2 dose regimens (higher or lower) of RP-G28 or placebo in a 1:1:1 ratio via a centralized, randomized Interactive Response (IXR) system using the subject ID number assigned at the beginning of screening. The IXR system was then used to assign drug kit numbers.

Subjects were asked to refrain from ingesting lactose-containing beverages/foods during the treatment period and record LI symptoms and adverse events (AEs) on a daily basis. The first dose of the study drug was administered on day 1 and the final dose on day 30. Subjects were followed for an additional 30 days after the final dose, during which time subjects were encouraged to ingest lactose-containing foods and record symptoms of LI daily. The demographics and characteristics of the study population are provided in Table 2.

### 2.3. Treatment Dose and Period

The lower dose of RP-G28 was 5 grams twice daily on days 1–10 followed by 7.5 grams twice daily on days 11–30. The higher dose of RP-G28 was 7.5 g twice daily for days 1–10, followed by 10 g twice daily on days 11–30. The placebo (powdered corn syrup that matched the consistency, color, sweetness, and taste of the drug) was administered in a blinded matching packet.

### 2.4. At 30 Days Post-Treatment “Real-World” Observation Period

The amount of lactose consumed each day for 7 days pre-treatment, and for 30 days post-treatment, was assessed via a food diary. Further, during the 30 day post-treatment period, global patient assessment questionnaires measured feelings, experiences and dietary changes resulting from treatment (Table 3). Global assessments are widely accepted as qualitative tools to evaluate the efficacy of treatment for functional disorders of the gastrointestinal tract [10].

### 2.5. Gut Microbiome Analysis

The 16S rRNA amplicon sequencing was performed utilizing patient stool samples collected on days 0, 31 and 61. Amplification using universal primers targeting the V4 region of the bacterial 16S rRNA gene was performed on 12.5 nanograms of total DNA from collected samples. ^10^ Each 16S rRNA gene amplicon was purified using the AMPure XP reagent (Beckman Coulter, Indianapolis, IN). Next, each sample was amplified using a limited cycle Polymerase Chain Reaction (PCR) program, adding Illumina sequencing adapters and dual-index barcodes (index 1(i7) and index 2(i5)) (Illumina) to the amplicon target. The final libraries were again purified using the AMPure XP reagent, quantified and normalized prior to pooling. The DNA library pool was then denatured with NaOH, diluted with hybridization buffer and heat denatured before loading on the MiSeq reagent cartridge and on the MiSeq instrument (Illumina). Automated cluster generation and paired-end sequencing with dual reads were performed according to the manufacturer’s instructions.

The 1050 DNA samples corresponding to 345 subjects receiving placebo, lower-dose or higher-dose treatments at three time points (days 0, 31, and 61) were analyzed by high-throughput quantitative PCR (qPCR) targeting *Bifidobacteria* and *Lactobacilli* using specific 16S rRNA gene and GroEL probes [11,12]. Microfluidic qPCR was performed using a BioMark HD reader (Fluidigm Corporation, San Francisco, CA) with a Dynamic Array 24.192 chip processed following manufacturer’s instructions.

### 2.6. Statistical Analysis

#### 2.6.1. Primary Efficacy Endpoint

A LI Symptom Questionnaire was developed, validated and applied during this clinical trial. This questionnaire rated individual symptoms of LI on an 11-point Numerical Response Scale (NRS) as well as a Verbal Rating Scale (VRS).

The primary efficacy endpoint was the proportion of responders on day 31. A responder was defined as a patient with a reduction from baseline in the composite score of 4 points or greater or a composite score of 0 (i.e., symptom resolution) on day 31. The composite score was calculated from the maximum symptom scores for each symptom (abdominal pain, cramping, bloating, and gas movement) after a lactose challenge test. Each symptom was rated on a 10-point Likert-type scale over 5 h after a lactose challenge. Each maximum symptom was then averaged into a composite score ranging from 0 to 10, where 0 indicated no symptoms and 10 indicated symptoms at their worst. A 4-point change was considered a meaningful improvement in symptoms of LI, based on psychometric analysis (blinded review of the clinical data prior to unblinding) and two rounds of cognitive interviews (N = 30 and 23). In the cognitive interviews, a 4-point change was meaningful for 84% of the subject responses and 53% of the subject responses in the first and second round of interviews, respectively. Symptom resolution was also considered meaningful. The 4-point threshold was further supported by empirical cumulative distribution functions using patient global severity anchor and was associated with sensitivity of 71%, specificity of 68%, positive predictive value of 63%, and negative predictive of 75%.

Statistical analysis used two-tailed tests at the α = 0.05 level of significance. The proportions of responders on day 31 were analyzed by a stratified (by quartile of baseline LI symptom composite score) Cochran–Mantel–Haenszel (CMH) test comparing both doses combined versus placebo, higher dose versus placebo, and lower dose versus placebo. Prior to unblinding, the protocol was modified to specify that the primary endpoint was a comparison of the 2 active arms combined versus placebo, using a two-sided test at the α= 0.05 level of significance. Pooling the data provided greater statistical power. The modified intent to treat (mITT) analysis population was all randomized subjects who received at least 1 dose of drug.

The sample size of 372 subjects was designed with the standard deviation of percentage abdominal pain reduction being 50%. It was hypothesized that the mean percentage reduction reported would be 50% for placebo-treated subjects and 70% for actively treated subjects. A simulation using Dunnett comparisons indicated that with 113 evaluable subjects in each arm, there would be 90% power to detect at least 1 of the active arms to be superior to placebo. The primary endpoint was based on a dichotomization of the distribution of the composite symptom score, which was thought to have at least as much power as percentage abdominal pain reduction.

The SAS software version 9.4 was used.

#### 2.6.2. Secondary Efficacy Endpoints

A number of secondary analyses were conducted. The proportion of participants in each group reporting no symptoms by the lactose intolerance composite score as well as the individual symptoms of abdominal pain, cramping, bloating, and gas movement after lactose challenge on day 31 was determined. The amount of milk consumed by participants on day 61 after RP-G28 or placebo was assessed. Global endpoints as defined in Table 3 were also compared on day 61 after RP-G28 or placebo. Analysis was performed by a stratified Cochran–Mantel–Haenszel (CMH) test for 3 comparisons: both doses combined versus placebo, higher dose versus placebo, and lower dose versus placebo. Analyses were conducted using two-sided tests at the α = 0.05 level of significance.

### 2.7. Gut Microbiome Analysis

Paired-end fastq files were joined into a single multiplexed, single-end fastq using the software tool fastq-join. Demultiplexing and quality filtering were performed on the joined results. Quality analysis reports were produced using the FastQC software [13]. Bioinformatics analysis of bacterial 16S amplicon sequencing data was conducted using the Quantitative Insights into Microbial Ecology (QIIME) software at a 25,000 reads/sample depth [14]. Operational Taxonomic Unit (OTU) picking was performed on the quality filtered results using pick_de_novo_otus.py. Chimeric sequences were detected and removed using ChimeraSlayer [15]. Summary reports of taxonomic assignment by sample and all categories were produced using QIIME summarize_taxa_through_plots.py and summarize_otu_by_cat.py. The script group_significance.py was used to compare taxa frequencies in sample groups and to determine whether there were statistically significant differences between abundances in the different groups. The non-parametric Analysis of Variance (ANOVA) test (Kruskal–Wallis) with False Discovery Rate (FDR) correction was used to compare treatments and placebo groups. For high-throughput qPCR data, the relative proportion of *Bifidobacterium* and *Lactobacillus* species was computed based on the Livak method. Quantitation cycle (Cq) values for each sample were normalized against the Cq value for the universal primers. Fold differences were calculated by 2 ^–ΔΔCt^. Paired t-test and ANOVA with Tukey tests were conducted to assess statistically significant differences between groups.

### 2.8. Safety

All treated subjects were included in the safety analyses using summary statistics by treatment group of AEs, concomitant medications, vital signs, physical examinations, and clinical laboratory measurements. AEs were reported from initiation of treatment through 30 days of the post-treatment period. An AE was classified as a Severe Adverse Event if it interfered significantly with the patients’ usual functions.

### 2.9. Irregular Site

A for-cause audit of one study site (out of 15 sites) was conducted due to significant data irregularities. The audit found significant deviations from the protocol and failure to comply with regulatory requirements for Good Clinical Practice (GCP), and identified protocol deviations including medical history and concomitant medications, inclusion/exclusion entry criteria, and administration of the Hydrogen Breath Test (HBT), lactose challenge, and patient diaries. Compared to the other sites, there were significant differences in multiple baseline symptom scores at this site. Patients reported two times more dairy intake prior to entry into the trial compared to subjects at other centers (*p* = 0.04). In addition, patients reported higher symptom severity scores during the in-clinic lactose challenge compared to the other site’s subjects (*p* = 0.035). High milk consumption in addition to high symptom severity scores based on a blinded in-clinic lactose challenge is inconsistent with patients with LI. Further, the screen failure rate was also significantly lower at this center compared to other centers.

Thus, efficacy analyses were conducted for both the mITT population which included all randomized subjects that received at least one dose of the drug and a mITT Efficacy Subset population, which included all randomized subjects that received at least one dose of the drug, excluding those who were enrolled at the site with numerous GCP violations.

## 3. Results

A total of 1398 subjects were screened for LI, 377 subjects from 15 study sites were enrolled and randomized (127 lower dose, 123 higher dose and 127 placebo), and 344 (87% placebo, 92% lower dose, 94% higher dose) completed the study. A high screen failure rate (>70%) was expected based on the rigorous inclusion and exclusion criteria required, including testing positive for LNP from a HBT and meeting LI symptom thresholds (63% of screen failures were due to HBT or LI symptoms scores not being met).

### 3.1. Primary Endpoint Analysis

#### Symptom Reduction in RP-G28-Treated Patients

In the Efficacy Subset mITT group, significantly more patients in the pooled RP-G28 group responded as compared to patients in the placebo group—40% versus 26% (*p* = 0.016). In total, 41% of subjects treated with lower-dose RP-G28 (*p* = 0.043) and 38% of subjects treated with higher-dose RP-G28 (*p* = 0.029) responded. (Table 4) In the mITT population, the pooled RP-G28 group trended toward significance *p* = 0.062 (40% with treatment versus 31% with placebo).

### 3.2. Secondary Endpoint Analysis

#### Individual Assessed Symptom Response to RP-G28 Post-Treatment

In the Efficacy Subset mITT, RP-G28-treated patients were significantly more likely than patients treated with placebo to report complete elimination of LI symptoms. RP-G28 was more likely to lead to complete elimination of the LI symptom composite score (*p* = 0.004) and individual symptoms of abdominal pain (*p* = 0.014), abdominal cramping (*p* = 0.002), abdominal bloating (*p* = 0.015), and gas movement (*p* = 0.001) than placebo (Figure 2A).

Patients treated with RP-G28 exhibited a consistent and significantly greater decrease in LI symptoms including cramping (*p* = 0.026) and bloating (*p* = 0.028). Non-significant trends for improvement were seen for abdominal pain (*p* = 0.105) and gas movement (*p* = 0.060) (Figure 2B).

“Real-world” Observation Period—Milk and Dairy Intake 30 days Post-Treatment: After 30 days post-treatment (day 61), patients treated with RP-G28 reported drinking significantly more milk, drinking an average of 1.5 cups/day versus 0.2 cups/day prior to treatment. The mean increase in milk intake of 1.3 cups/day (SD 1.479) in treated patients was significantly higher than that of the placebo group, which was 0.7 cups/day (SD 1.591) (*p* = 0.008, Efficacy Subset mITT) (Figure 2C).

In addition, 59% of treatment patients consumed ≥1 cups/d of milk after being treated with RP-G28 in comparison to 42% with placebo (*p* = 0.01, Efficacy Subset mITT)). Patients treated with RP-G28 also consumed more dairy in general, ingesting 5.4 cups/day on day 61 versus only 1 cup prior to treatment, at baseline. In comparison, the placebo group ingested 4.5 cups/day on day 61 versus 1.7 cups/day at baseline. The mean increase in dairy intake of 4.3 cups/day trended toward significance compared to the placebo group (*p* = 0.057, Efficacy Subset mITT).

“Real-world” Observation Period—Global Patient Assessments 30 days Post-Treatment: Significantly more treated patients (82%) reported “no symptoms” or “mild symptoms”, respectively) as compared to 64% in the placebo group after treatment (*p* = 0.001, Efficacy Subset mITT). In addition, significantly more treated patients reported satisfaction with the ability of RP-G28 to prevent or treat their LI symptoms, with 66% of treatment patients reporting “very satisfied” or “extremely satisfied” as compared to 52% in the placebo group (*p* = 0.030, Efficacy Subset mITT). Patients’ perception of adequate relief from LI symptoms and patients’ global impression of change were also improved with treatment (*p* = 0.042 and P = 0.034 respectively, Efficacy Subset mITT) (Figure 3).

### 3.3. Gut Microbiome Analysis

Of 543 bacterial species identified overall, 28 were differentially represented (Kruskal–Wallis, FDR corrected *p* < 0.05). Of those, relative abundances of five *Bifidobacterium* taxa (*Bifidobacterium_Other*, *Bifidobacterium sp*., *Bifidobacterium adolescentis*, *Bifidobacterium longum* and *Bifidobacterium pseudolongum*) were increased by both RP-G28 treatments. High-throughput qPCR quantitative data confirmed a significant (*p* < 0.05) increase in the abundance of the phylum Actinobacteria, the family *Bifidobacteriaceae*, and the genus *Bifidobacterium* on day 31 in the higher- and lower-dose treatment groups, but not the placebo group (Figure 4). Specifically, treatment with RP-G28 resulted in an elevated abundance of *Bifidobacterium longum*, *Bifidobacterium bifidum*, *Bifidobacterium breve*, *Bifidobacterium catenulatum*, *Bifidobacterium angulatum*, *Bifidobacterium gallicum*. In total, 78% (77/99) of patients treated with RP-G28 had elevated *Bifidobacteria* levels, as compared to 52% (49/94) of patients in the placebo group (*p* < 0.001). Data using Firmicutes, *Lactobacillaceae* and *Lactobacillus* species-specific 16S rRNA gene probes showed a significant increase in the relative abundance of the family *Lactobacillaceae* on day 31 in the lower-dose treatment and a non-significant increase in the higher-dose group, but not in the placebo group.

On day 31 after treatment with RP-G28, qPCR quantitative data revealed an increase in the relative abundance of the genus *Bifidobacterium* in the lower-dose galacto-oligosaccharide RP-G28 and higher-dose RP-G28 treatment groups, an effect not seen in the placebo group.

### 3.4. Safety

RP-G28 was well tolerated. In total, 6.9% of patients had treatment-emergent adverse events (TEAEs) with no differences between placebo and RP-G28. None of the serious adverse events (SAEs) were treatment related. No TEAEs resulted in hospitalization or death (Table 5).

## 4. Discussion

Though lactose intolerance is remarkably common, treatment options have not changed for decades [16,17]. The cornerstone of treatment, abstinence from consuming dairy products, is potentially nutritionally detrimental, inconvenient and limiting to patients. This study assessed a novel treatment strategy for lactose intolerance involving supplementation of RP-G28, which promoted the proliferation of lactose-fermenting bacteria, including *Lactobacillus* and *Bifidobacterium* [12]. Increased abundance and a higher diversity of these species is associated with enhanced levels of beta-galactosidase [17], reduced lactose-derived gas production in the colon [9,17], and mitigation of clinical symptoms in patients with LI [12,18].

To assess the efficacy and meaningfulness of treatment, three broad groups of LI patient experiences were evaluated—a LI composite score to measure specific symptoms, milk and dairy intake to evaluate patients’ ability to increase dietary lactose with treatment, and global assessment measures to assess patients’ overall satisfaction with therapy. This three-pronged assessment provided a comprehensive evaluation of how treatment influenced symptoms of LI and the day to day lives of patients.

A composite symptom score and a stringent threshold of meaningful change was established prior to unblinding the study in order to identify patients who reported a meaningful decrease in LI symptoms. Psychometric analyses indicated that the four symptom severity items were distinct yet related enough to support the creation of a composite score, and this composite score was shown to be reliable, valid, and responsive to change in LI symptoms over time. This tool provides a valuable resource for accurately assessing symptoms from the patient perspective in future LI clinical trials, and for identifying subjects who have experienced a meaningful treatment benefit in terms of patient-reported LI symptom severity. This study is one of the first to define and measure a meaningful treatment benefit for LI patients. The tool was developed adhering to good measurement principles following the FDA PRO requirements [19], and with consultation with the FDA. The instrument content was developed through a literature review, patient surveys, and concept elicitation and was tested through several rounds of cognitive interviews.

In the Efficacy Subset mITT population, the primary endpoint of response on day 31 was achieved in the pooled and individual RP-G28 groups. Individual symptoms were consistently reduced. The most rigorous endpoint for symptom relief is complete elimination. In the Efficacy Subset mITT population, RP-G28 administration resulted in complete symptom relief in a significantly greater proportion of patients than placebo, for the measures of abdominal pain, cramping, bloating, gas movement and overall symptoms (*p* < 0.05).

RP-G28-treated patients drank significantly more milk after treatment compared to the control group, drinking an average of 1.3 cups/day, as compared to an average of 0.7 cups/day more by the placebo group. Milk is the primary source of lactose in the diet, and thus milk consumption is an important indicator of the clinical benefits of supplementation with RP-G28. The ability to drink milk without LI symptoms supports optimal nutrient intake, contributing to the USDA’s recommended 2–3 cups of dairy per day [20]. The sustained improvement in dairy intake and symptom relief on day 61 are further validation of the durability of treatment.

Consistent, statistically significant improvement in global assessments support the clinical meaningfulness of RP-G28. When consuming dairy foods for 30 days after treatment, 82% of patients reported “no or mild symptoms”, 66% reported being “very or extremely satisfied,” 83% reported adequate relief and 40% reported “very much or much improvement”.

Overall, RP-G28 was safe and well tolerated. There were no differences in adverse events in those receiving RP-G28 or placebo. There were no serious adverse events in either group. Of note, there were no increases in GI side effects with RP-G28. This is important as RP-G28 is a GOS and, thus, part of the family of fermentable, oligo, di, monosaccharides, and polyols referred to as FODMAPs. FODMAPs have been shown to trigger GI symptoms in patients with irritable bowel syndrome. It is reassuring that at the doses administered, RP-G28 did not induce any significant GI side effects.

Gut microbiome changes and a reduction in net hydrogen gas production support the hypothesized mode of action [18]. In total, 78% of individuals treated with RP-G28 had elevated *Bifidobacteria*, and *Lactobacillaceae*. Patient microbiomes adapted further with the reintroduction of lactose into patient diets during a 30 day period post-treatment, with an increase in lactose-fermenting *Roseburia* species [2,12]. Elevated abundance of *Bifidobacterium* and *Lactobacillus* resulting from RP-G28 treatment, known to enhance lactose fermentation, is likely instrumental in better clinical outcomes. Further, stool from patients adapted to lactose produce less hydrogen, due to an absolute reduction in hydrogen production, rather than an alteration in hydrogen uptake by the microbiome [21]. Additional studies to understand how the changes in gut microbiome induced by RP-G28 lead to clinical improvements are warranted.

## 5. Limitations

One of the 15 study sites was excluded from the Efficacy Subset mITT due to significant GCP violations. Nevertheless, this is the largest and most rigorously designed double-blinded randomized trial for the treatment of LI ever conducted. Another limitation is that the construct that identifies LI is subjective, depending on a single lactose dose and a subsequent time course. Real-world LI is likely intermittent, and depends on diet, dose, transit and other environmental and biological factors that are impossible to fully control within a clinical trial. However, in this study, the primary endpoint is part of a comprehensive assessment, including primary and secondary efficacy endpoints, global patient assessments, lactose consumption assessments, and correlation of treatment to changes in the microbiome—all of which are indicative of a beneficial effect of RP-G28 in improving symptoms of LI.

Another limitation of the study is the duration, which does not allow determination of long-term durability. While significant improvements in diet quality and lactose tolerance were evident on days 31 and 61, we cannot determine whether retreatment will be necessary or effective.

## 6. Summary

RP-G28 was safe and effective for reducing or eliminating symptoms of LI. Treatment with RP-G28 led to increased milk and dairy intake and improved quality of life. RP-G28 was safe and well tolerated at the doses administered. RP-G28 led to changes in the microbiome, which may be involved in the clinical benefits observed in patients with LI. These findings are relevant not only to lactose intolerance, but also usher in an era of using prebiotics to manipulate the microbiome to facilitate gut health.

## Figures and Tables

**Figure 1 nutrients-12-01058-f001:**
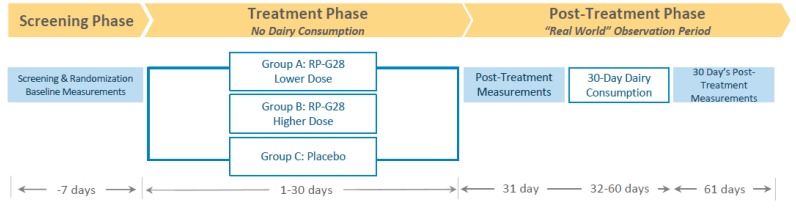
Protocol Design.

**Figure 2 nutrients-12-01058-f002:**
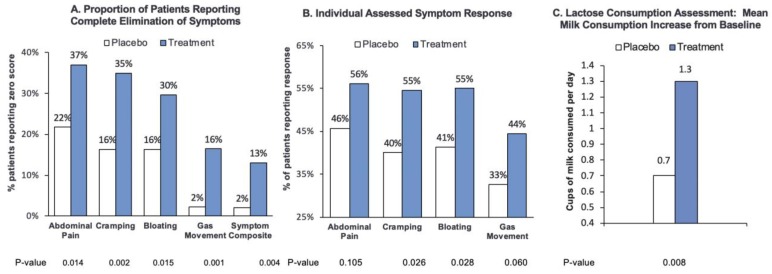
Secondary Endpoints Analysis. (**A**) The proportion of patients reporting complete elimination of lactose intolerance (LI) symptoms (abdominal pain, cramping, bloating, and gas movement) with RP-G28 treatment or placebo. (**B**) The proportion of patients reporting a response (≥4-point improvement from baseline or a score of 0 on day 31) in each key LI symptom (abdominal pain, cramping, bloating, and gas movement) with RP-G28 treatment or placebo. (**C**) RP-G28 led to a significantly greater increase in daily average milk consumption from baseline 30 days after treatment (day 61) compared to placebo.

**Figure 3 nutrients-12-01058-f003:**
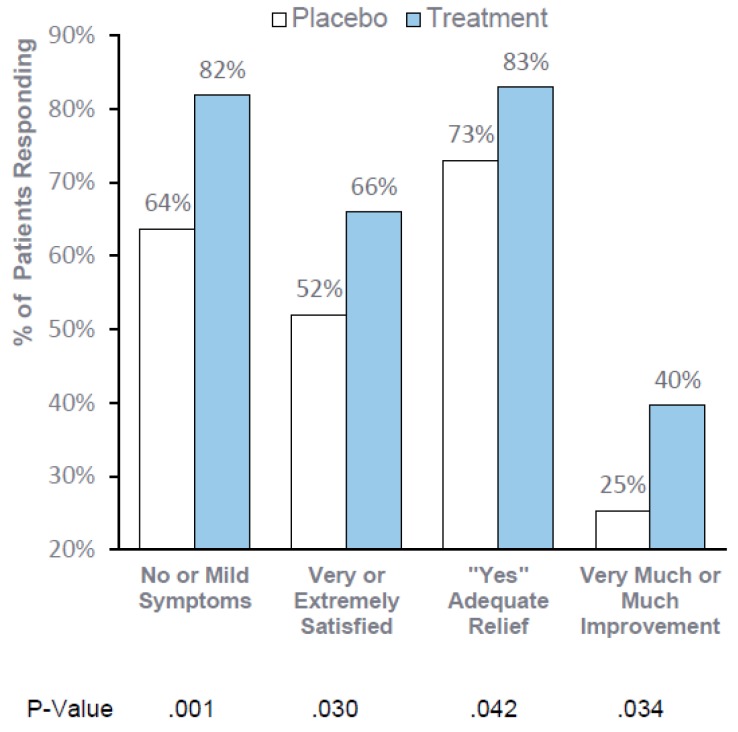
Global Patient Assessments. Patients’ global assessments were evaluated 30 days after treatment (day 61) and were based on a patient questionnaire.

**Figure 4 nutrients-12-01058-f004:**
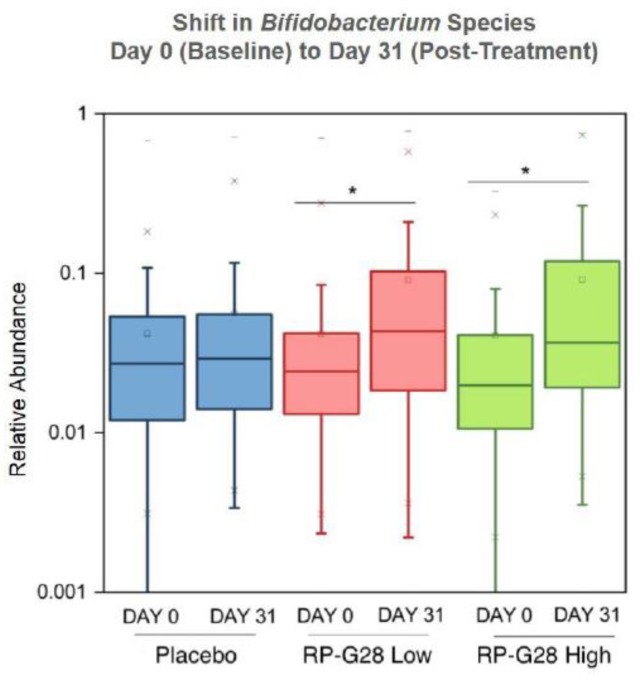
Gut Microbiome Analysis.

**Table 1 nutrients-12-01058-t001:** Diagnosis and Main Criteria for Inclusion.

Male or female subjects, and female subjects were to be non-pregnant, and non-lactating.
Aged 18 to 75 years, inclusive.
A Hydrogen Breath Test (HBT) result that was positive for lactase deficiency.
A total abdominal pain score of at least 5 and at least 1 time point rated 3 or higher on the 11-point Numerical Rating Scale (NRS) over the 5 h screening.
At least 2 individual symptom scores present, measured over a 5 h screening: abdominal cramping, bloating, movement of gas (stomach rumbling), release of gas (flatulence), and bowel urgency.

**Table 2 nutrients-12-01058-t002:** Demographics and Baseline Characteristics (mITT Population).

Characteristic	Placebo (*N* = 121)	Lower Dose (*N* = 126)	Higher Dose (*N* = 121)	All Randomized (*N* = 368)
Age (year)				
Mean (SD)	39.9 (13.03)	42.8 (12.75)	40.8 (12.96)	41.2 (12.93)
Median	37	45	40	41
Min, Max	19, 74	18, 73	18, 70	18, 74
Gender (N [%])				
Male	43 (35.5)	48 (38.1)	57 (47.1)	148 (40.2)
Female	78 (64.5)	78 (61.9)	64 (52.9)	220 (59.8)
Ethnicity/Race (N [%])				
*Hispanic or Latino*				
African American	0	4 (3.2)	5 (4.1)	9 (2.4)
American Indian or Alaska Native	0	1 (0.8)	0	1 (0.3)
Asian	1 (0.8)	0	0	1 (0.3)
White	42 (34.7)	48 (38.1)	33 (27.3)	123 (33.4)
Other	0	3 (2.4)	1 (0.8)	4 (1.1)
*Not Hispanic or Latino*				
African American	55 (45.5)	51 (40.5)	57 (47.1)	163 (44.3)
American Indian or Alaska Native	1 (0.8)	1 (0.8)	2 (1.7)	4 (1.1)
Asian	7 (5.8)	5 (4.0)	2 (1.7)	14 (3.8)
White	15 (12.4)	13 (10.3)	17 (13.5)	45 (12.2)
Other	0	0	4 (3.3)	4 (1.1)
Height (cm)				
Mean (SD)	167.0 (8.97)	168.5 (8.82)	170.0 (10.60)	168.5 (9.54)
Median	167	168	168.9	167.8
Min, Max	147, 188	147, 194	146, 199	146, 199
Weight (kg)				
Mean (SD)	82.3 (21.97)	87.7 (24.28)	86.3 (19.08)	85.4(21.91)
Median	77.2	85.3	85.4	83.1
Min, Max	41, 166	51, 163	45, 138	41, 166
BMI (kg/m)				
Mean (SD)	29.5 (7.82)	30.8 (8.09)	29.9 (6.37)	30.1 (7.48)
Median	27.2	29.6	29.3	29
Min, Max	18, 58	17, 57	18, 50	17, 58

BMI = body mass index; max = maximum; min = minimum; mITT = modified intent to treat; *N* = number of subjects; SD = standard deviation; kg = kilogram.

**Table 3 nutrients-12-01058-t003:** Global Patient Assessment Tool.

Assessment Questionnaires (Quality of Life Instrument)	Response Type	Scale	Description
Patient Global Impression of Severity	NRS^1^	5-point	1 = no symptoms, 1 = mild, 2 = moderate, 3 = severe, 4 = very severe
Patient Assessment of Satisfaction	NRS	5-point	1 = not at all satisfied, 2 = a little satisfied, 3 = somewhat satisfied, 4 = very satisfied, 5 = extremely satisfied
Patient Assessment of Adequate Relief	Binary	2-point	Yes/No
Patient Global Impression of Change	Likert-type scale	7-point	1 = very much improved, 2 = much improved, 3 = minimally improved, 4 = no change, 5 = minimally worse, 6 = much worse, 7 = very much worse

1. NRS = Numerical Rating Scale.

**Table 4 nutrients-12-01058-t004:** Primary Endpoint ^1^

Efficacy Subset mITT ^2^ (N = 296)			
	Higher dose	Lower dose	Pooled (Higher + Lower)
Number of subjects	97	102	199
RP-G28 Treatment	37 (38%)	42 (41%)	79 (40%)
Placebo	25 (26%)	25 (26%)	25 (26%)
CMH *p*-value versus placebo ^4^	0.029	0.043	0.016
mITT ^3^ (N = 368)			
	Higher dose	Lower dose	Pooled (Higher + Lower)
Number of subjects	121	126	247
RP-G28 treatment	46 (38%)	53 (42%)	99 (40%)
Placebo	38 (31%)	38 (31%)	38 (31%)
CMH *p*-value versus placebo ^4^	0.096	0.117	0.062

1. Proportion of LI symptom composite score responders post-treatment (day 31). 2. Efficacy Subset (mITT)—mITT data in which observed inconsistent data from one study center was removed from analysis. 3. mITT—modified intent to treat (all patients who received at least one dose of drug). 4. CMH = Cochran–Mantel–Haenszel. *p*-value versus placebo. N = number of subjects.

**Table 5 nutrients-12-01058-t005:** Overall Summary of Treatment-Emergent Adverse Events (Safety Population).

Type of Adverse Event	Placebo (N = 121)	Lower Dose (N = 126)	Higher Dose (N = 121)	Pooled Dose (Lower + Higher) (N = 247)
Subject with at least 1 TEAE	38 (31.4%)	40 (31.7%)	36 (29.8%)	76 (30.8%)
Subjects with at least 1 treatment-related TEAE ^1^	8 (6.6%)	11 (8.7%)	6 (5.0%)	17 (6.9%)
Subjects with at least 1 SAE ^2^	3 (2.5%)	1 (0.8%)	0	1 (0.4%)
Subjects with at least 1 treatment-related SAE	0	0	0	0
Subjects with a TEAE leading to study drug withdrawal, interruption, or reduction	2 (1.7%)	1 (0.8%)	0	1 (0.4%)
Any AE resulting in death	0	0	0	0

AE = adverse event; N = Number of subjects; SAE = Serious adverse event; TEAE = treatment-emergent adverse event. 1. Treatment-related TEAT was defined as any TEAT that was possibly, probably, or definitely related to study drug. 2. One subject (a randomized placebo) experienced the SAE of spontaneous abortion.

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
