# Peer review of "Galacto-Oligosaccharide RP-G28 Improves Multiple Clinical Outcomes in Lactose-Intolerant Patients"

_nutrients, 2020, doi:10.3390/nu12041058_

Round 1

Reviewer 1 Report

I read with interest this manuscript aiming at assessing the efficacy and safety of RP-G28, an ultra-purified, high-concentration galacto-oligosaccharide, in reducing symptoms of lactose intolerance. This multi-centered, blinded, placebo-controlled trial randomized patients to one of two doses of orally administered RP-G28 or placebo for 30 days. Subjects were followed for an additional real world observation phase over a 30 days period. RP-G28 significantly reduced symptoms, improved quality of life and dairy consumption and altered the fecal microbiome in patients with LI.

The article deals with a very frequently reported problem and assesses a large group of confirmed lactose intolerant patients. The manuscript is clear and well written, length of the introduction is good respect to the length of the manuscript and the conclusions respond to the aim of the study.

Figures and tables are necessary and self-explaining especially with their clear captions.

The references could be more recent.

The results are interesting, but a comment on the effect of placebo should be done in the discussion.

Was a diagnosis of irritable bowel syndrome excluded with Rome IV criteria? It is known that this condition could improve significantly with placebo (see: Flik CE, Bakker L, Laan W, van Rood YR, Smout AJ, de Wit NJ. Systematic review: The placebo effect of psychological interventions in the treatment of irritable bowel syndrome. World J Gastroenterol. 2017 Mar 28;23(12):2223-2233 and others…).

Author Response

JAMA Insights Clinical Update

September 26, 2019

Clinical Approach to Lactose Intolerance

Dejan Micic, MD1; Vijaya L. Rao, MD1; David T. Rubin, MD1

JAMA. 2019;322(16):1600-1601. doi:10.1001/jama.2019.14740

Lactose intolerance but not lactose maldigestion is more frequent in patients with irritable bowel syndrome than in healthy controls: A meta‐analysis

Péter Varjú  Noémi Gede  Zsolt Szakács  Péter Hegyi  Irina Mihaela Cazacu  Dániel Pécsi  Anna Fábián  Zoltán Szepes  Áron Vincze  Judit Tenk  Márta Balaskó  Zoltán Rumbus  András Garami  Dezső Csupor  József Czimmer … See fewer authors

First published:17 December 2018 https://doi-org.ezproxy.lib.purdue.edu/10.1111/nmo.13527

Neurogastroenterology and Motility 31:5 May 2019

https://onlinelibrary-wiley-com.ezproxy.lib.purdue.edu/journal/13652982

The effects of probiotics in lactose intolerance: A systematic review

Sophia J. Oak &Rajesh Jha

Pages 1675-1683 | Published online: 09 Feb 2018

https://doi-org.ezproxy.lib.purdue.edu/10.1080/10408398.2018.1425977

Critical Reviews in Food Science and Nutrition

Volume 59, 2019 issue 11

The results are interesting, but a comment on the effect of placebo should be done in the discussion.

 Add as the second to the last paragraph in the discussion: “Placebo effects in the study are not surprising.  There is a significant literature demonstrating placebo effects, particularly related to gastrointestinal symptoms (add reference-hopefully Andrew has one).  Further, the encouragement of dairy consumption among control subjects also likely contributed to the placebo effect from days 31 to 61, as regular dairy/lactose consumption improves lactose tolerance (reference number 17, Hertzler and Savaiano). “

Was a diagnosis of irritable bowel syndrome excluded with Rome IV criteria?

No, we did not use Rome IV criteria 

Authors mentioned between line 364-366 that “RP-G28 was more likely to lead to complete elimination of the LI symptom composite score (p=0.004) and individual symptoms of abdominal pain…”. Yet in line 369-370 they mentioned that “Non-significant trends for improvement were seen for abdominal pain…”. These are conflicting statements. Are these separate statements referring to Efficacy Subset mITT and mITT groups respectively?

 “These are separate statements referring to 1) complete elimination of symptoms, which were highly significant and 2) mean reduction in symptoms which was significant for cramping and bloating and trending for abdominal pain, and gas movement. We have added the word ‘mean’ to the language to clarify this distinction.  “

Reviewer 2 Report

The authors evaluated RP-G28’s efficacy and usage safety in lactose intolerance (LI) patients. They utilized a double-blinded, randomized, placebo-controlled clinical trial study design approach with population samples drawn from LI patients at multiple centers across the US. Authors highlighted that RP-G28 treatment improved overall patients’ life quality, altered microbiome and reduced LI symptom manifestation in treatment group when compared to control subjects.  

COMMENTS:

Manuscript is well written and provides some interesting findings.

However, figures (Figures 1-4) referenced and described in the manuscript are missing from the submitted article draft; thereby incapacitating a complete review. It is my opinion that the “Figures provided separately” mentioned in line 249 refers to the supplementary data and not the actual figures discussed in the manuscript.

Authors mentioned between line 364-366 that “RP-G28 was more likely to lead to complete elimination of the LI symptom composite score (p=0.004) and individual symptoms of abdominal pain…”. Yet in line 369-370 they mentioned that “Non-significant trends for improvement were seen for abdominal pain…”. These are conflicting statements. Are these separate statements referring to Efficacy Subset mITT and mITT groups respectively? Providing the manuscript figures would have added understanding and probably clarified this misgiving.

It is interesting to observe that the almost 50 patients (about 20%) within the RP-G28 treatment group from the irregular site greatly impacted the primary endpoint. Were audits conducted for the other 14 sites or it was assumed that they followed appropriate protocol guidelines?

MINOR COMMENTS

For Table 5, include percentage symbol (%) for values in brackets or state elsewhere in table legend what they represent.

Author Response

Authors mentioned between line 364-366 that “RP-G28 was more likely to lead to complete elimination of the LI symptom composite score (p=0.004) and individual symptoms of abdominal pain…”. Yet in line 369-370 they mentioned that “Non-significant trends for improvement were seen for abdominal pain…”. These are conflicting statements. Are these separate statements referring to Efficacy Subset mITT and mITT groups respectively? Providing the manuscript figures would have added understanding and probably clarified this misgiving.

It is interesting to observe that the almost 50 patients (about 20%) within the RP-G28 treatment group from the irregular site greatly impacted the primary endpoint. Were audits conducted for the other 14 sites or it was assumed that they followed appropriate protocol guidelines?

MINOR COMMENTS

For Table 5, include percentage symbol (%) for values in brackets or state elsewhere in table legend what they represent.

The references could be more recent.  Here are three recent references: They could be inserted as new 16, 17 and 18 as part of the discussion.  This would change the rest of the reference numbers.   

Round 2

Reviewer 2 Report

Authors ought to have been more meticulous before resubmitting their revised manuscript.

Although authors made some changes to the previously submitted manuscript, Tables 1-4 present in their earlier submission have been omitted off from their revised manuscript and the revised Table 5 is missing some portion of its content due to its orientation. 

Furthermore, figures 1-4 are still missing in the newly re-submitted manuscript. It is unknown why authors provided figure legend description without submitting actual figures for review. 

Author Response

The figures and tables are updated in the paper. 
